# Prenatal Particulate Matter (PM) Exposure and Natriuretic Peptides in Newborns from Mexico City

**DOI:** 10.3390/ijerph18126546

**Published:** 2021-06-18

**Authors:** Abigail Muñoz-Rodríguez, Jorge Alfonso Maciel-Ruiz, Ana María Salazar, Monserrat Sordo, Patricia Ostrosky-Wegman, Jorge H. Limón-Pacheco, Andrés Eduardo Nepomuceno-Hernández, Rodrigo Ayala-Yáñez, María Eugenia Gonsebatt, Citlalli Osorio-Yáñez

**Affiliations:** 1Departamento de Medicina Genómica y Toxicología Ambiental, Instituto de Investigaciones Biomédicas, Universidad Nacional Autónoma de México (UNAM), Ciudad Universitaria, Apartado Postal 70228, Ciudad de México 04510, Mexico; abigail.munozrodriguez@gmail.com (A.M.-R.); anamsm@biomedicas.unam.mx (A.M.S.); monserratsordo@hotmail.com (M.S.); patricia.ostrosky@gmail.com (P.O.-W.); jorgehlimon@comunidad.unam.mx (J.H.L.-P.); mgonsebatt@gmail.com (M.E.G.); 2Instituto Nacional de Cancerología (INCan), Subdirección de Investigación Básica, Ciudad de México 14080, Mexico; jmaciel.lp@gmail.com; 3Centro de Investigación Materno Infantil del Grupo de Estudios al Nacimiento, Asociación Hispano Mexicana, Ciudad de México 09880, Mexico; nepomucenoandres@gmail.com (A.E.N.-H.); rayalaabc@gmail.com (R.A.-Y.)

**Keywords:** air pollution, particulate matter exposure, pregnancy, natriuretic peptides, newborns, ANP, BNP, CNP

## Abstract

(1) Background: The aim of this study was to assess associations between particulate matter (PM) exposure and natriuretic peptide concentrations in cord blood from newborns. (2) Methods: we conducted a cross-sectional study in Mexico City with 101 pregnant women from CIMIGEN Hospital. Atrial natriuretic peptide (ANP), B-type natriuretic peptide (BNP) and C-type natriuretic peptide (CNP) were measured in plasma from cord blood in 51 newborns by ELISA. We estimated PM exposure (PM2.5 and PM10) at first, second and third trimester of pregnancy. (3) Results: The median and interquartile range for ANP, BNP and CNP plasma concentrations were 66.71 (46.92–80.23), 98.23 (73.64–112.30) and 1129.11 (944.10–1452.02) pg/mL, respectively. PM2.5 and PM10 levels for the whole pregnancy period were 22.2 µg/m^3^ and 41.63 µg/m^3^, respectively. Employing multivariable linear regression models adjusted for maternal age, newborn sex, smoking before pregnancy, maternal occupation and newborns’ length and height, we observed a 2.47 pg/mL (95%CI: −4.67, −0.27) decrease in BNP associated with PM2.5 exposure during second trimester. Adjusted for the same set of confounders, third trimester PM10 exposure was inversely associated with ANP concentrations (beta estimate: −0.90; 95% CI: −1.80, −0.03). Neither PM10 nor PM2.5 were associated with CNP at any trimester of pregnancy. (4) Conclusions: Prenatal exposure to particulate matter was associated with ANP and BNP decrease in newborns.

## 1. Introduction

Ambient air pollution is a global health concern that more profoundly impacts people from low- and middle-income countries [1]. Mexico City is one of the main Latin American megacities and registers poor air quality with particulate matter (PM) concentrations above the national and international standards [2,3,4]. Among all ambient air pollutants, particulate matter less than 2.5 µm (PM2.5) is of public health concern because of the small particulate size is able to reach the alveolus and distributes to other organs and systems, exerting adverse health effects [5].

Pregnant women and newborns are a susceptible population to the adverse health effects of air pollution. However, studies linking ambient air pollution and perinatal outcomes have been conducted mainly in Caucasian or Asian populations. For example, ambient air pollution has been associated with higher risk of preeclampsia [6], gestational diabetes [7], pregnancy loss [8], low birth weight [9], thyroid dysfunction [10] and increased blood pressure in newborns [11]. Moreover, scarce evidence exists linking PM2.5 exposure and cardiovascular outcomes in newborns. For instance, van Rossem et al. (2015) showed positive associations between late pregnancy exposure to PM2.5 and black carbon with systolic blood pressure in newborns [11]. Similarly, Madhloum et al. (2019) found associations between PM2.5 exposure (each at a 5 µg/m^3^ increment) and higher systolic (2.4 mmHg; 95% CI: 0.5 to 4.2) and diastolic (1.8 mmHg; 95% CI: 0.2 to 3.5 mmHg) blood pressure at birth [12]. Several neurohormonal mechanisms regulate blood pressure homeostasis including the renin-angiotensin-aldosterone system (RAAS) [13], antidiuretic hormone [13], nitric oxide [14,15], endothelin-1 [14] and natriuretic peptides [16], among others. The natriuretic peptide family includes three members: atrial natriuretic peptide (ANP), brain natriuretic peptide (BNP) and C-type natriuretic peptide (CNP) [16]. Natriuretic peptides are hormones that participate in cardiac remodeling, blood pressure and volume balance [17]. ANP and BNP are synthetized in the atria and ventricles of the heart, respectively. CNP is produced mainly by endothelial cells and the brain [18,19]. Natriuretic peptides are secreted in response to mechanical stress produced by volume or pressure overload of the myocardial walls; then, once secreted, they induce diuretic, natriuretic and vasodilatory effects in order to maintain cardio renal homeostasis. Natriuretic peptides have been employed as prognostic and diagnostic markers of cardiovascular disease, heart failure and stroke, also in apparently healthy adults [18]. However, natriuretic peptides concentrations and clinical significance in newborns have been less studied. For example, a study previously reported higher BNP concentrations in newborns from pre-eclamptic women and newborns exposed to stress conditions such as complications during labor and delivery [20]. Higher levels of BNP have been reported in neonates of mothers with type 1 diabetes and antenatal stress [21]. In addition, an increase in maternal plasma aminoterminal proCNP (NTproCNP) concentrations and ANP polymorphisms (22,387 > C) have been linked to adverse obstetric events and preeclampsia, respectively [22,23]. Overall, previous studies have shown that natriuretic peptides are biomarkers of stress conditions in pregnant women (complicated delivery, preeclampsia, type 1 diabetes) [20,21,22,23] and might be one of the mechanisms underlying the link between PM2.5 exposure and hypertension risk in newborns. Therefore, the aim of this study was to assess associations between PM exposure (PM2.5 and PM10) during pregnancy and natriuretic peptide concentrations in cord blood of newborns from Mexico City.

## 2. Materials and Methods

### 2.1. Study Participants

The characteristics of the study were previously described in Sordo et al. (2019) [24]. Briefly, two hundred and forty women in their last quarter of pregnancy who attended CIMIGEN (Centro de Investigación Materno Infantil del Grupo de Estudios al Nacimiento) for prenatal care were invited to participate in this study during the period of October 2016 to April 2018. The objectives of the study were explained to the volunteers who signed an informed consent form and completed a structured questionnaire with information regarding age, parity, residence area, occupation, general health condition, medications, smoking habits and exposure to genotoxic agents, such as X-rays. A total of 110 women met the inclusion criteria. All women included in the study took iron and folic acid as part of their prenatal care and all newborns were born to term (gestational age at delivery more or equal to 36 weeks). Exclusion criteria included living near gasoline stations, paint and plastic factories or solvents, active smoking during pregnancy, occupational exposure to polycyclic aromatic hydrocarbons (PAHs), women with premature rupture of membranes and any illness during the last three months of pregnancy. The study protocol was approved by the Ethics Committee of the Instituto de Investigaciones Biomédicas, Universidad Nacional Autónoma de México and CIMIGEN, Mexico City. After birth, blood samples were collected immediately from umbilical cord blood into tubes containing heparin and then transported on ice until they arrived at the laboratory. Samples were centrifugated to obtain plasma and frozen at −80 °C until analyses.

### 2.2. Environmental Monitoring

Environmental exposure to PM2.5 and PM10 was estimated using 24 h real-time data from the Automatic Atmospheric Monitoring Network (RAMA, for its Spanish initials) stations closest to the home of participants as previously described [24]. Briefly, according to the date of delivery, the average exposure levels of PM2.5 and PM10 of the first, second and third trimester of pregnancy were estimated.

### 2.3. Cotinine Measurements

Cotinine concentrations in urine were measured employing a semi-quantitative test using a One-Step Cotinine Test Device (Certum Diagnostic, Nuevo Leon, Monterrey, Mexico), according to manufacturer’s protocol. Cotinine concentrations in urine higher than or equal to 200 ng/mL were considered positive or indicative of active smoking.

### 2.4. Natriuretic Peptide Concentrations in Newborns

ANP, BNP and CNP were measured in the plasma samples by enzyme-linked immunosorbent assays (ELISA) according to the manufacturer’s instructions (My BioSource, San Diego, CA, USA). Natriuretic peptides were measured in technical duplicate for each one (intra-assay coefficient variation <15%), and the average was recorded as the mean concentrations in pg/mL for ANP, BNP and CNP.

### 2.5. Statistical Analyses

We performed exploratory analyses to assess data quality and consistency. Shapiro-Wilk test statistics were employed to check normality of the continuous variables. All continuous variables are described as a geometric median and interquartile range. Frequencies and percentages were reported for categorical variables. We analyzed correlations between PM exposure and natriuretic peptides using Spearman correlation coefficients because PM exposure variables were not normally distributed. Then, we assessed associations between trimester-specific PM exposure and natriuretic peptides employing a cross-sectional approach and a longitudinal approach. In the cross-sectional approach, we performed multivariable linear regression analyses to assess the associations between PM exposure at each trimester of pregnancy and natriuretic peptides (ANP, BNP or CNP) adjusted for covariates selected by the forward method; the selection was based on biological plausibility, their influence on the model fit, or their effect on the association between the PM2.5 exposure and natriuretic peptides. In addition to the cross-sectional approach, we performed linear mixed models to assess the mean effect of PM2.5 measured at the first, second and third trimester of pregnancy on newborns’ natriuretic peptides. Mixed-effects models take into account the correlation between repeated measurements on subjects over time [25]. Trimester of pregnancy was considered as a random effect in the model, and the remaining independent variables were considered as fixed effects. All models were adjusted for the same set of confounders: maternal age (years, continuous), smoking before pregnancy, newborns’ sex (male, female), maternal occupation (yes, no), newborns´ length and height. We report the beta estimate and 95% confidence intervals (CIs) for each model. Statistical analyses were performed using RStudio software 3.3.1 (R Foundation for Statistical Computing, Vienna, Austria). All reported P-values are two sided and deemed significant at alpha = 0.05.

## 3. Results

### 3.1. Characteristics of the Study Participants, PM Levels and Natriuretic Peptide Concentrations

Of the initial 110 mother–newborn pairs, only 51 newborns had enough cord blood samples for BNP and CNP and 48 for ANP measurements. Table 1 shows maternal and newborn characteristics and prenatal exposure to PM2.5 and PM10. For those mothers included in this study, median maternal age was 25 years old with an interquartile range from 23 to 30 years; 37.3% reported to have an occupation during pregnancy and 21.6% reported to have smoked before pregnancy. Newborns included in this study had a median height and weight of 3200 g and 50 cm, respectively. For those included, 53% of the newborns were boys and had a median Capurro value of 39. Overall, PM10 exposure was higher than PM2.5. The median concentrations for PM2.5 for those included pairs and for the whole pregnancy period was 22.2 µg/m^3^ and for PM10 was 41.63 µg/m^3^. Both PM2.5 and PM10.0 were above the WHO air quality guideline annual values of 10 µg/m^3^ and 20 µg/m^3^, respectively [26]. The percentage of mothers who worked during pregnancy was lower in mothers included than mothers excluded from this study. On the contrary, the percentage of mothers who smoked before pregnancy was higher in mothers included than excluded. There were no differences in newborn characteristics for those included and excluded from the study. PM2.5 exposure was higher for the first and second trimester in those excluded than included in the study. PM10 exposure was significantly higher during first trimester in those excluded than those included in the study.

Table 2 shows natriuretic peptide concentrations in newborns. The median and interquartile range for ANP, BNP and CNP were 66.71 (46.92–80.23), 98.23 (73.64–112.30) and 1129.11 (944.10–1452.02) pg/mL, respectively.

### 3.2. Correlations between Natriuretic Peptides, Characteristics of the Study Population and Particulate Matter Exposure

Natriuretic peptides were not correlated with maternal age or newborn characteristics such as weight and height (Table 3). ANP, BNP or CNP were not different according to maternal occupation or smoking status before pregnancy (data not shown). We observed higher ANP concentrations in girls compared to boys (Median and IQR: 75.75 (58.91–93.96) pg/mL and 53.60 (45.52–70.96), respectively), while no differences were found for BNP or CNP according to newborns’ sex (data not shown). At the second trimester, PM2.5 and PM10 were marginal correlated with lower BNP concentrations ((r = −0.252; *p*-value = 0.075) and (r = −0.27; *p*-value = 0.054), respectively). ANP was significantly correlated with PM2.5 at second (r = 0.30; *p* = 0.037) and third trimester (r = −0.32; *p* = 0.025) of pregnancy. Similarly, ANP correlated with PM10 at second (r = 0.26, *p* = 0.07) and third trimester (r = −0.39; *p*-value = 0.01), but only the third trimester reached statistical significance. Overall, CNP was not significantly correlated with PM2.5 or PM10 at any time point during pregnancy (Table 3).

Figure 1 shows Spearman correlation coefficients among natriuretic peptides, and we observed significant correlation between ANP and CNP (r = 0.34; *p* = 0.020). No significant correlations were observed for ANP and BNP or BNP and CNP.

### 3.3. Multivariable Models for Associations between Natriuretic Peptides and PM2.5 or PM10 Exposure during Pregnancy

Based on multivariable regression models adjusted for maternal age (years, continuous), newborns´ sex and smoking before pregnancy, PM2.5 exposure during second trimester was significantly associated with a decrease in plasma BNP concentrations in newborns. This association remained statistically significant even after adjustment for maternal occupation and newborn height and weight. For instance, we observed a 2.47 pg/mL (95% CI: −4.67, −0.27) decrease in BNP associated with PM2.5 exposure during second trimester (Model 4, Table 4). PM2.5 exposure during first and third trimester were not significantly associated with newborns´ BNP concentrations (Table 4). Similarly, we found statistically significant associations between PM10 exposure at the second trimester and BNP concentrations in newborns (Table 5). No associations were observed between PM10 exposure during the first and third trimester and BNP levels. We observed statistically significant associations between third trimester PM10 exposure and ANP (beta estimate: −0.90 and 95%; CI: −1.80, −0.03) in multivariate models adjusted for maternal age, sex, smoking before pregnancy, maternal occupation, newborn weight and height (Table 5). Neither PM2.5 nor PM10 exposure at any time point during pregnancy was associated with CNP concentrations (Table 4 and Table 5).

### 3.4. Linear Mixed Models to Assess Associations between Natriuretic Peptides and PM2.5 and PM10 Exposure during Pregnancy

Results from linear mixed models adjusted for maternal age, newborns´ sex, smoking before pregnancy, maternal occupation, newborns´ height and length showed that PM2.5 concentrations during pregnancy were not statistically significantly associated with ANP (beta estimate and 95% CI: −0.018 (−1.12, 1.08); *p*-Value = 0.97); BNP (beta estimate and 95% CI: −0.099 (−1.09, 0.89); *p*-Value = 0.84) or CNP concentrations (beta estimate and 95% CI: −6.31 (−41.08, 28.46); *p*-Value = 0.72). Similar results were obtained when we assessed associations between PM10 and natriuretic peptides (ANP, BNP or CNP) employing linear mixed models adjusted for the same set of confounders.

## 4. Discussion

To the best of our knowledge, this study is the first to report associations between prenatal exposure to particulate matter and newborns´ natriuretic peptides. The main finding of our study was the decreased in ANP and BNP concentrations associated with PM10 (third trimester) and PM2.5 (second trimester) exposure, respectively. We did not observe associations between CNP and PM2.5 or PM10 exposure at any time point during pregnancy. These results suggest that prenatal ambient air pollution, particularly PMs, might decrease natriuretic peptide concentrations in newborns, and further studies should confirm our findings and evaluate the mechanisms underlying these associations.

Natriuretic peptides such as BNP have been employed as biomarkers of neonatal stress such as delivery mode [27], low birth weight [28], uterine contraction and antenatal stress [29]. Previous reports showed high concentrations of BNP in newborns (day 0) and then BNP levels decrease gradually until they reach a steady state [30].

BNP plasmatic concentrations from cord blood at day 0 (98.09 pg/mL) from our study were higher compared to BNP serum concentrations in 3-day-old full-term or preterm newborns (55.1 pg/mL and 25.5 pg/mL, respectively) [31]. On the other hand, BNP concentrations in our study were lower than those reported in cord blood for mothers with gestational diabetes (114 +/− 39.0 pg/mL.) [32].

There are few epidemiologic studies assessing associations between BNP concentrations and ambient air pollution and all of them have been conducted in adults. For instance, PM2.5 was not significantly associated with whole blood BNP concentrations (0.8% increase (95% CI: −16.4, 21.5; *p*-Value = 0.94)) in patients with chronic obstructive disease [33]. Another study suggested that BNP may be used as a biomarker of systolic and diastolic right ventricle dysfunction in women exposed to biomass fuel [34]. On the other hand, Caravedo and colleagues (2014) found no associations between chronic exposure to biomass fuel smoke and NT-proBNP levels in men and women living in a high-altitude setting [35]. Results from an experimental study with Albino Wistar rats exposed sub-chronically to environmental tobacco smoke showed an increase in mRNA expression of hypertrophic genes such as BNP, ß-myosin heavy chain and ANP compared to the control group [36].

Notably, PM2.5 exposure during second trimester was significantly associated with lower BNP concentrations at birth. Natriuretic peptides appear to be functional by midgestation and regulate blood pressure, salt and water balance in the developing embryo. Natriuretic peptides also act as vasodilators in the placenta vasculature, and peaks in ANP/BNP expression during gestation suggest a role of natriuretic peptides in cardiac angiogenesis [37]. The link between prenatal PM2.5 and BNP concentrations in newborns may suggest cardiotoxic or hypertensive effects of PM2.5 in early life. BNP has both adverse and protective effects depending on the stage of the disease. For example, in severe disease stages, BNP is a biomarker of cardiac hypertrophy related to high blood pressure, or volume overload due to BNP becomes resistant and is no longer able to compensate. Contrary, in the early stages of disease progression, BNP has protective effects by inducing natriuresis and diuresis and counteracting the unwanted effects of activated RAAS and sympathetic nervous system (SNS) in order to restore blood pressure levels [38,39,40]. Therefore, BNP represents a relevant protective mechanism toward the development of hypertension [16]. Future studies should be conducted to evaluate whether a decrease in BNP concentrations is linked to a hypertension risk in newborns.

ANP concentrations in our study measured at day 0 (median 66.71 pg/mL) were lower than those measured at day 1.4 in 31 preterm infants with idiopathic respiratory disease 620 pg/mL [41]. To note, we observed inverse associations between PM10 exposure at the third trimester and ANP concentrations. It is difficult to explain the reason why we observed associations with PM10 but not with PM2.5 and ANP in our study. We hypothesized that the reason might be due to this population being exposed to higher concentrations of PM10 than PM2.5 (median concentrations at third trimester 44.75 vs. 24.65 µg/m^3^) or probably differences in chemical composition for PM2.5 and PM10 may explain this association (PM10 and ANP). A study conducted in Mexico City found differences in chemical composition for PM2.5 and PM10. For PM2.5, organic matter was the major component (48.9%), inorganic aerosols were the second major component, accounting for 26% (18.4% (NH_4_)_2_SO_4_ and 7.6% NH_4_NO_3_) and elemental carbon accounted for 17%. For PM10, geological material was the major component accounting for 37.4%, elemental carbon was 9.5% and nitrates were 6.1% [42]. It is difficult to compare our results with other studies linking ambient air pollution and ANP levels because none were conducted in newborns. For example, an epidemiologic study showed that ANP concentrations were higher, but not statistically significant, in traffic police officers compared with controls (officers with indoor activities) [43]. We are aware of only one experimental study showing that motorcycle exhaust increased left ventricular mass and ANP mRNA levels in rats [44].

To the best of our knowledge, no previous studies, either epidemiologic or experimental, have evaluated the link between PM2.5 exposure and C-type natriuretic peptides. CNP has vasodilatory effects and PM2.5 has been associated with hypertension in newborns [11,45]. Thus, we hypothesized a decrease in CNP concentrations associated with prenatal exposure to CNP. However, we observed overall no associations between PM2.5 exposure at any trimester and CNP in newborns. Therefore, future studies should be conducted to confirm our findings.

The results of our study should be interpreted in light of its strengths and limitations. The strengths of our study include PM2.5 exposure measurement during pregnancy and natriuretic peptide measurement in cord blood. The major limitation of our study is its observational nature; therefore, residual or unmeasured confounding cannot be completely ruled out. Another limitation of our study is the small sample size and the fact that we do not have measurements of blood pressure in the newborns to assess the link between PM2.5, blood pressure and natriuretic peptides. Due to the cross-sectional design of our study, we cannot assume a causal connection between PM2.5 levels and BNP. Despite all these limitations, ours is the first study to assess cardiac risk biomarkers in relation to ambient air pollution in newborns.

## 5. Conclusions

Prenatal exposure to particulate matter is associated with a decrease in ANP and BNP in newborns. Our results highlighted the importance of reducing PM emission to reduce adverse health effects in susceptible populations such as pregnant women and newborns. Our findings should be confirmed in other study populations, and additional research is needed to determine the consequences of natriuretic peptides decrease associated with PMs exposure later in life.

## Figures and Tables

**Figure 1 ijerph-18-06546-f001:**
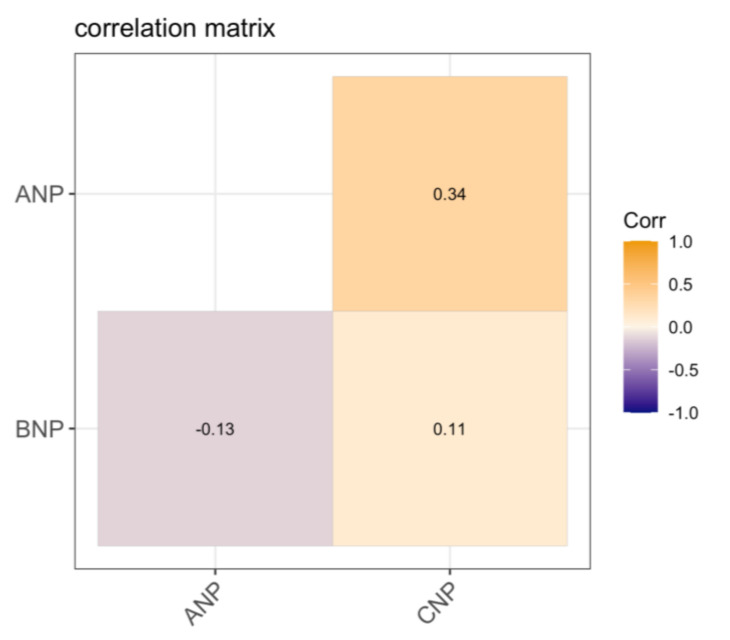
ANP, atrial natriuretic peptide; BNP, B-type natriuretic peptide; CNP, C-type natriuretic peptide. Values represent spearman correlation coefficients. Only the ANP and CNP correlation was statistically significant (*p* = 0.020). No significant correlations (*p* > 0.05) were found for ANP and BNP or BNP and CNP.

**Table 1 ijerph-18-06546-t001:** Maternal and newborn characteristics in 110 participants from CIMIGEN Clinic in Mexico City.

Characteristics	Total (*n* = 110)	Included (*n* = 51)	Excluded (*n* = 59)	*p*-Value
Maternal characteristics				
Age (years)	27.00 (23.75–31.25)	25.00 (23.00–30.00)	29.00 (25.00–32.00)	0.06
Ocupation				
Yes	52 (47.27)	19 (37.26)	33(55.93)	<0.001
No	50 (45.46)	32 (62.75)	18 (30.51)	
Smoking before pregnancy				
Yes	19 (17.3)	11 (21.57)	8 (1.36)	0.579
No	84 (76.36)	40 (78.43)	44 (74.6)	
Newborn characteristics				
Weight (g)	3120 (2880–3450)	3200 (3000–3460)	3050 (2700–3450)	0.19
Height (cm)	50.0 (49.00–51.50)	50.00 (49.00–51.00)	50.00 (49.00–52.00)	0.86
Sex				
Girls	53 (48.18)	23 (45.10)	30 (50.85)	0.71
Boys	54 (49.09)	27 (52.94)	27 (45.76)	
Capurro	39.00 (38.00–40.00)	39.00 (39.00–40.00)	39.00 (38.00–40.00)	0.062
PM2.5 exposure (μg/m^3^)				
First trimester	22.47 (18.79–25.49)	20.15 (16.42–24.53)	23.63 (19.86–27.50)	<0.001
Second trimester	23.32 (18.73–26.65)	22.55 (17.36–24.90)	24.13 (20.49–26.88)	0.018
Third trimester	24.65 (18.39–27.74)	24.85 (18.41–29.33)	24.63 (18.47–26.55)	0.40
Whole Pregnancy	22.63 (21.07–24.71)	22.16 (20.37–22.74)	24.43 (22.00–25.64)	<0.001
PM10 exposure (μg/m^3^)				
First trimester	39.26 (33.06–50.63)	35.00 (29.35–47.14)	45.85 (34.89–54.73)	<0.001
Second trimester	44.77 (34.14–49.41)	42.44 (33.19–48.69)	46.66 (34.55–54.30)	0.19
Third trimester	44.73 (32.97–56.88)	47.35 (32.22–57.11)	42.25 (35.04–55.22)	0.93
Whole pregnancy	42.58 (39.44–47.66)	41.63 (38.44–43.10)	46.63 (40.21–48.79)	0.002

Data represents median and interquartile range (IQR) or N (%). *p* values derived from Wilcoxon or chi-squared test. Missing values ranged from 1 for Capurro or Newborns´ weight and height to 8 for maternal occupation.

**Table 2 ijerph-18-06546-t002:** Natriuretic peptide concentrations in plasma from cord blood.

	ANP (pg/mL)	BNP (pg/mL)	CNP (pg/mL)
N	48	51	51
Mean (Min–Max)	67.78 (1.54–258.7)	98.09 (39.23–279.60)	1475 (5.76–7344.73)
Median and IQR	66.71 (46.92–80.23)	98.23 (73.64–112.30)	1129.11 (944.10–1452.02)

Abbreviations: IQR, interquartile range; ANP, atrial natriuretic peptide; BNP, brain natriuretic peptide; CNP, C-type natriuretic peptide.

**Table 3 ijerph-18-06546-t003:** Spearman’s correlations between plasma natriuretic peptides, maternal and newborn characteristics and particulate matter exposure.

Variables	ANP (pg/mL)		BNP (pg/mL)		CNP (pg/mL)	
	r	P	r	P	r	P
Maternal characteristics						
Age (years)	0.04	0.81	−0.03	0.85	−0.05	0.72
Newborn characteristics						
Weight (g)	−0.16	0.28	0.19	0.17	0.08	0.58
Height (cm)	−0.11	0.45	0.23	0.11	0.12	0.40
Capurro	−0.14	0.33	−5.1 × 10^–4^	0.99	−0.14	0.33
PM2.5 exposure (μg/m^3^)						
First trimester	0.10	0.48	0.11	0.44	−0.08	0.58
Second trimester	0.30	0.04	*−0.25*	*0.08*	−0.01	0.94
Third trimester	−0.32	0.03	0.11	0.43	0.07	0.61
Whole pregnancy	−0.04	0.80	−1.7 × 10^−3^	0.99	−0.03	0.86
PM10 exposure (μg/m^3^)						
First trimester	0.17	0.24	0.12	0.41	−0.06	0.68
Second trimester	*0.26*	*0.07*	*−0.27*	*0.05*	0.07	0.62
Third trimester	−0.39	0.01	0.11	0.46	0.03	0.84
Whole pregnancy	−0.004	0.98	−0.13	0.35	0.09	0.55

Abbreviations: PM2.5, particulate matter exposure less than 2.5 µm; PM10, particulate matter exposure less than 10 µm; ANP, atrial natriuretic peptide; BNP, brain natriuretic peptide; CNP, C-type natriuretic peptide. Significant results (*p* < 0.05) are highlighted in bold and marginally significant results (*p* < 0.10) are given in italics.

**Table 4 ijerph-18-06546-t004:** Linear regression models to assess associations between PM2.5 exposure and newborns’ natriuretic peptides.

	PM_2.5_ First Trimester	PM_2.5_ Second Trimester	PM_2.5_ Third Trimester	PM_2.5_ Whole Pregnancy
BNP (pg/mL)	β (95% Cl)	β (95% Cl)	β (95% Cl)	β (95% Cl)
Model 1	1.05 (−1.17, 3.26)	**−2.59 (−4.82, −0.36)**	0.66 (−0.95, 2.28)	−0.06 (−5.89, 5.77)
Model 2	0.97 (−1.29, 3.23)	**−2.76 (−5.02, −0.50)**	0.603 (−1.05, 2.25)	−1.05 (−7.61, 5.52)
Model 3	0.70 (−1.51, 2.90)	**−2.65 (−4.83, −0.47)**	0.53 (−1.06, 2.12)	−1.94 (−8.31, 4.42)
Model 4	0.58 (−1.62, 2.77)	**−2.47 (−4.67, −0.27)**	0.63 (−0.95, 2.21)	−1.23 (−7.65, 5.18)
CNP (pg/mL)				
Model 1	5.65 (−69.2, 80.50)	2.27 (−76.65, 81.20)	−1.47 (−55.86, 52.93)	11.16 (−184.03, 206.34)
Model 2	−3.27 (−77.96, 71.42)	−7.07 (−85.77, 71.63)	−8.22 (−62.5, 46.03)	−69.83 (−284.14, 144.54)
Model 3	−3.02 (−79.12, 73.09)	−7.21 (−86.90, 72.48)	−8.16 (−63.09, 46.76)	−70.04 (−288.69, 148.60)
Model 4	−2.03 (−79.21, 75.16)	−9.30 (−90.71, 72.12)	−8.99 (−64.70, 46.72)	−78.40 (−302.49, 145.69)
ANP (pg/mL)				
Model 1	0.19 (−2.33, 2.70)	2.14 (−0.44, 4.72)	−1.32 (−3.02, 0.39)	−1.43 (−8.08, 5.22)
Model 2	0.30 (−2.26, 2.87)	2.27 (−0.33, 4.88)	−1.27 (−3.03, 0.49)	−0.64 (−8.40, 7.08)
Model 3	0.45 (−2.08, 2.99)	2.13 (−0.47, 4.72)	−1.20 (−2.94, 0.54)	−0.25 (−7.88, 7.39)
Model 4	0.47 (−2.14, 3.07)	2.15 (−0.49, 4.78)	−1.22 (−3.0, 0.55)	−0.25 (−8.01, 7.52)

Model 1. Adjusted for maternal age, newborn sex, smoking before pregnancy. Model 2. Model 1 + occupation. Model 3. Model 1 + occupation + newborn weight. Model 4. Model 1 + occupation + newborn weight + newborn height. Significant results (*p* < 0.05) are highlighted in bold.

**Table 5 ijerph-18-06546-t005:** Linear regression models to assess associations between PM10 exposure and newborns’ natriuretic peptides.

	PM_10_ First Trimester	PM_10_ Second Trimester	PM_10_ Third Trimester	PM_10_ Whole Pregnancy
BNP (pg/mL)	β (95% Cl)	β (95% Cl)	β (95% Cl)	β (95% Cl)
Model 1	0.23 (−0.74, 1.21)	**−1.32 (−2.46,−0.17)**	0.30 (−0.52, 1.11)	−0.48 (−3.06, 2.09)
Model 2	0.20 (−0.79, 1.19)	**−1.43 (−2.60, −0.27)**	0.27 (−0.57, 1.10)	−0.94 (−3.76, 1.88)
Model 3	0.06 (−0.91, 1.03)	**−1.41 (−2.53, −0.29)**	0.20 (−0.60, 1.01)	−1.59 (−4.34, 1.16)
Model 4	0.03 (−0.93, 0.99)	**−1.30 (−2.44, −0.17)**	0.24 (−0.56, 1.04)	−1.29 (−4.07, 1.50)
CNP (pg/mL)				
Model 1	12.15 (−20.4, 44.65)	8.11 (−32.29, 48.52)	−3.51 (−30.95, 23.93)	28.93 (−56.94, 114.80)
Model 2	9.0 (−23.41, 41.42)	2.48 (−38.12, 43.07)	−6.68 (−33.97, 20.60)	3.12 (−89.6, 95.9)
Model 3	9.33 (−23.8, 42.43)	2.45 (−38.62, 43.52)	−6.65 (−34.33, 21.03)	3.91 (−91.9, 99.72)
Model 4	9.57 (−23.9, 43.04)	1.29 (−40.93, 43.51)	−6.99 (−35.03, 21.04)	1.16 (−97.3, 96.4)
ANP (pg/mL)				
Model 1	0.52 (−0.63, 1.70)	0.93 (−0.39, 2.24)	−0.96 (−1.80, −0.12)	−0.96 (−3.96, 2.03)
Model 2	0.56 (−0.60, 1.73)	1.01 (−0.33, 2.35)	−0.95 (−1.81, −0.08)	−0.73 (−4.07, 2.62)
Model 3	0.65 (−0.49, 1.80)	0.96 (−0.36, 2.28)	−0.90 (−1.76, −0.04)	−0.35 (−3.70, 2.99)
Model 4	0.67 (−0.51, 1.84)	0.98 (−0.37, 2.34)	−0.90 (−1.80, −0.03)	−0.36 (−3.80, 3.05)

Model 1. Adjusted for maternal age, newborn sex, smoking before pregnancy. Model 2. Model 1 + occupation. Model 3. Model 1 + occupation + newborn weight. Model 4. Model 1+ occupation + newborn weight + newborn height. Significant results (*p* < 0.05) are highlighted in bold.

## Data Availability

Data available on request due to restrictions. The data presented in this study are available on request from the corresponding author. The data are not publicly available due to privacy restrictions.

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
