# Peer review of "Prenatal Particulate Matter (PM) Exposure and Natriuretic Peptides in Newborns from Mexico City"

_ijerph, 2021, doi:10.3390/ijerph18126546_

Round 1

Reviewer 1 Report

General

Please consider to review the text by a native speaker in English.

The idea of your study is intersting. Nevertheless, I suggest deducing the rationale for your study in their introduction a little more persuasive. Why are natriuretic peptides so interesting in new-borns? What will this biomarker stand for? Is there a causal connection? Please explain and discuss.

line 59: participates

line 61: is produced ... please add "and the brain" including a proper citation.

line 92: premature rupture of membranes 92 by an infectious process: Please explain.

line 286: Would you please introduce a hypothesis for further studies?

Author Response

REVIEWER 1

General

Please consider reviewing the text by a native speaker in English. 

The idea of your study is interesting. Nevertheless, I suggest deducing the rationale for your study in their introduction a little more persuasive. Why are natriuretic peptides so interesting in newborns? What will this biomarker stand for? Is there a causal connection? Please explain and discuss.

We thank the reviewer for the comments and suggestions that improve our manuscript.

Our manuscript will be reviewed by a native speaker before publication as part of the APC of the journal and we are sure that this revision will avoid any grammar mistakes in our document.

As suggested by the reviewer, we made the rationale of our study more persuasive indicating the meaning and utility of natriuretic peptides in newborns as follows:

Introduction, lines: 126-129.

“Overall, previous studies have shown that natriuretic peptides are biomarkers of stress conditions in pregnant women (e.g., complicated delivery and preeclampsia)(20-23) and might be linked to hypertension in newborns”.

We cannot assume a causal connection between air pollution and natriuretic peptide levels in newborns. In this version of the manuscript, we indicated that due to the cross-sectional design of our study, we can not assume causality. Please see lines 404-405 in the discussion section.

line 59: participates

Sorry, for this oversight

line 61: is produced ... please add "and the brain" including a proper citation.

The change was made as suggested.

line 92: premature rupture of membranes 92 by an infectious process: Please explain.

Women with premature rupture of membranes (PRM) for any reason were excluded from the study. We deleted “by an infectious process”.

line 286: Would you please introduce a hypothesis for further studies?

We added the hypothesis for future studies as suggested.  Lines 360-362.

We thank the reviewer for the careful review of our manuscript.

Reviewer 2 Report

This study is excellent, as it is the first to report associations between prenatal exposure to particulate matter and newborns´ natriuretic peptides. The results suggest that prenatal ambient air pollution, particularly PM might decrease natriuretic peptide concentrations in newborns.

Only one suggestions for revising: Please give the main results about PM exposure in the present study, as this is an important information in this work.

Author Response

We thank the reviewer for his/her comments.

We added the main results for PM exposure in the results section, please see lines 213-217.

Also, we added the main results in the abstract.

We thank the reviewer for the careful review of our manuscript.